# School Built Environments and Bullying Behaviour: A Conceptual Model Based on Qualitative Interviews

**DOI:** 10.3390/ijerph192315955

**Published:** 2022-11-30

**Authors:** Jacinta Francis, Gina Trapp, Natasha Pearce, Sharyn Burns, Donna Cross

**Affiliations:** 1Telethon Kids Institute, 15 Hospital Avenue, Nedlands, WA 6009, Australia; 2Centre for Child Health Research, The University of Western Australia, 35 Stirling Hwy, Nedlands, WA 6009, Australia; 3School of Population and Global Health, The University of Western Australia, 35 Stirling Hwy, Nedlands, WA 6009, Australia; 4School of Population Health, Curtin University, Kent St, Bentley, WA 6102, Australia

**Keywords:** bullying, peer victimisation, built environment, school, children, qualitative, conceptual model

## Abstract

Interest in how the school built environment impacts bullying behaviour has gained momentum in recent years. While numerous studies have identified locations within schools where bullying frequently occurs, few studies have investigated the potential conceptual pathways linking school locations to bullying behaviour. This study aimed to (i) identify school built environment factors that may prevent or facilitate bullying behaviour in primary and secondary schools; and (ii) develop a conceptual model of potential pathways between the school built environment and bullying behaviour for future anti-bullying intervention research. Seventy individual semi-structured interviews were conducted between May and December 2020, with policymakers (*n* = 22), school staff (*n* = 12), parents (*n* = 18), and students (*n* = 18). School staff, parents and students, were recruited from six metropolitan primary and secondary schools in Perth, Western Australia. Interviews were conducted online and face-to-face using semi-structured interview guides. A thematic analysis was undertaken. Participants identified school bullying locations (e.g., locker areas, bathrooms, corridors) and built environment factors linked to bullying behaviour via (i) visibility and supervision; (ii) physical and psychological comfort and safety; and (iii) social-emotional competencies. The findings have policy and practice implications regarding the design of school built environments to prevent bullying behaviour.

## 1. Introduction

Bullying is the repeated, intentional harm or humiliation of a person who has less power than the aggressor [1,2]. In Australia, meta-analyses of studies involving children and young people have indicated a lifetime prevalence for traditional bullying victimisation of 25% and perpetration of 12% [3]. Meta-analyses of international studies involving adolescents have estimated mean prevalence rates of 35% for traditional bullying victimisation and 15% for cyberbullying [4]. The consequences of bullying victimisation during childhood can include depression, anxiety, poor general health and suicidal ideation and behaviour [5,6,7,8,9]. In addition, involvement in childhood bullying has been associated with poor academic performance, criminal behaviour or delinquency, and adult unemployment [10,11,12].

School-based anti-bullying interventions typically target the whole-school community—including students, school staff and parents—and comprise multiple program components addressing school climate, policies, classroom rules, curriculum, teacher training, and parent engagement [13]. These multi-faceted interventions are often guided by social-ecological models that recognise the reciprocal influences on the bullying behaviour of individuals, peers, schools and broader community and societal contexts [14]. While school-based interventions have been shown to reduce bullying behaviour, the effectiveness of many anti-bullying interventions has been modest, with reductions in bullying perpetration of 19–20% and victimisation of 15–16% [13,15,16].

An awareness of the factors contributing to bullying behaviour is an important precursor to developing effective anti-bullying interventions. Despite evidence that school built environment factors may influence violence between students [17,18,19], few studies have provided comprehensive assessments of school built environment factors influencing bullying behaviour. However, research exploring bullying behaviour and the school built environment (i.e., the buildings, spaces and objects created or modified by people) has been gaining momentum in recent years. A 2022 scoping review found 43 studies had identified school bullying locations or ‘hotspots’, with an additional 19 studies identifying specific school built environment factors linked to bullying behaviour, such as security cameras, architectural design, aesthetics, seating, and vandalism [20]. Fram and Dickmann’s (2012) study of an elementary school in the United States identified school built environment factors that exacerbated bullying and peer harassment, including large playgrounds, poor lighting, large distances between buildings and playgrounds, insufficient windows overlooking playgrounds, long hallways, and isolating building designs [21]. The authors concluded that the school built environment could exacerbate existing tendencies within students to bully or harass peers [21]. Similarly, Horton and colleagues (2020) conducted a qualitative study of Swedish teachers’ perceptions of school spatial elements (structural, environmental, and social) influencing social relations between students and teachers’ ability to prevent school bullying [22]. Environmental elements influencing negative student interactions included school size and diversity within spaces, while elements influencing bullying prevention included size (both the number of students attending a school and the size of the physical space) and obstructive playground and building design [22].

Empirical investigations of pathways linking school built environment factors to bullying behaviour have not appeared in the published literature. However, studies have linked school built environment factors, such as obstructive building design, to poor supervision of students [21,22]. Bullying researchers have also noted bullying occurs more frequently in unstructured locations where supervision is lacking [2,23,24]. Similarly, a common strategy to prevent school bullying is to enhance students’ social-emotional competencies, including self-control, emotional regulation, empathy, negotiation, and communication skills [14,25]. Social time at school, particularly during break times, can provide students with opportunities to develop and apply skills that may prevent bullying [2]. Subsequently, playgrounds containing diverse play spaces and engaging equipment may assist students in developing the social skills needed to prevent bullying and facilitate positive interactions between students [22,26,27]. There have been calls, however, for further research into the role of school design and the built environment on bullying behaviour [22]. This study explored the perspectives of school policymakers, practitioners, staff, students, and parents to identify school built environment factors that prevent or facilitate bullying behaviour in primary and secondary schools. A secondary aim was to develop a conceptual model of potential pathways between the school built environment and bullying behaviour for future anti-bullying intervention research.

## 2. Materials and Methods

Ethics approval was granted by the Human Research Ethics Office at The University of Western Australia (RA/4/20/4995), and governance approval from Catholic Education Western Australia (CEWA) and the Department of Education WA.

### 2.1. Sampling and Recruitment

#### 2.1.1. Policy Makers and Practitioners

Policymakers and practitioners (hereafter referred to as policymakers) were invited to participate in individual interviews between May and December 2020. Policymakers included system-level decision makers, school architects, private consultants (e.g., disability services, landscape and playground design), and academics specialising in bullying prevention, education, mental health, and Aboriginal and Torres Strait Islander health. Policymakers were recruited using a snowball sampling technique after consultation with the research team’s Stakeholder Advisory Committee, comprising representatives from Western Australian school sectors, government departments, school associations, private architectural firms, universities and child health research institutes.

#### 2.1.2. School Staff, Students, and Parents

Staff from Perth metropolitan schools were recruited and interviewed between August and December 2020. Maximum variation sampling was used to recruit schools from different school sectors, socio-economic areas, years of establishment, and school levels (i.e., primary or secondary). Six schools were recruited, with schools evenly divided between (i) independent, Catholic and government; (ii) primary (i.e., students aged 4 to 12) and secondary (students aged 12 to 18); (iii) low and high socio-economic areas; and iv) old and new school builds, with “old” schools built prior to 1979. Socio-economic status was determined using the Index of Community Socio-Educational Advantage (ICSEA), with ICSEA scores <1000 representing low socio-economic schools and scores ≥1000 representing high socio-economic schools [28].

School principals were contacted via email to ascertain their interest in the study. Consenting principals nominated a study coordinator to assist the research team in recruiting school staff, students, and parents for the individual semi-structured interviews. School study coordinators recruited two school staff from each school, including principals and deputy principals, classroom teachers, specialist subject teachers, chaplains, and school psychologists. School study coordinators purposively recruited two students and two parents for each student age category (grades 4–6, 7–8, 9–10) and were asked to nominate participants with different experiences of bullying (i.e., targets of bullying, perpetrators of bullying, and students with no bullying experience), however, the interviewer was not made aware of individual student bullying status unless stated by the participant during the interview. Once participants returned their signed consent forms, the research team organised a suitable interview day, time, and location. Written consent was provided by all participants.

### 2.2. Data Collection

Semi-structured interview discussion guides were developed by a team of researchers with expertise in school design, built environments, bullying, mental health and school-based interventions. Separate discussion guides were prepared for (i) policymakers; (ii) primary school children; (iii) secondary school children; (iv) school staff and parents. The interview discussion guides were piloted with six people to determine face validity, with only minor changes required. Policymakers, staff, and students were asked to describe key issues influencing bullying behaviour in school students, while students were asked to describe a time they saw or experienced bullying behaviour at school. These descriptions included the identification of bullying locations or “hotspots”. All participants were asked to describe (i) what changes were needed to reduce bullying behaviour in schools; (ii) what prevented people from making the suggested changes to reduce bullying behaviour; and (iii) what would help people make the suggested changes to reduce bullying behaviour. Participants were prompted to consider the built environment, social environment, policy environment and individual factors influencing bullying behaviour. School staff, students, and parents also referred to a map of their school depicting the layout of buildings and outdoor areas. Participants were prompted to look at the map when thinking about key issues influencing bullying behaviour or describing times they saw or experienced bullying incidents at school. Bullying was described by the interviewer using age-appropriate language as (i) a repeated behaviour; (ii) causing intentional harm or humiliation; and (iii) involving a power imbalance between the bullying target and aggressor.

All interviews were conducted by the first author, an experienced qualitative researcher. Study participants were given the option of a videoconference, face-to-face, or telephone interview. Face-to-face interviews were conducted at the participant’s workplace or school. A second researcher was present during all face-to-face interviews with children, with most student interviews conducted in person.

Interviews with policymakers and students ranged between 40 and 70 min and 20 and 30 min, respectively. Interviews with school staff and parents ranged between 20 and 45 min. All interviews were audio-recorded with participant permission. Data collection continued until no new codes or themes emerged from the data. A survey designed to capture demographic data was self-administered at face-to-face interviews and interviewer-administered during videoconference interviews.

### 2.3. Data Analysis

Dependability and credibility were maintained by ensuring the interview transcripts were checked by two researchers for accuracy and analysed with the assistance of the qualitative research software package QSR NVivo, Version 12 (QSR International, Burlington, MA, USA). A theoretical thematic analysis was conducted, with a coding framework based on the individual, social environment, built environment and policy factors underpinning the project’s social-ecological framework, the study aims and interview discussion guide. A critical realism perspective enabled the exploration of participants’ experiences while considering social-ecological influences [29].

Analysis was guided by Braun and Clarke’s (2006) framework for thematic analysis. Two research team members familiarised themselves with the data by re-reading transcripts and developing initial ideas and codes [29]. Open coding was used to assign the data to codes which were then grouped into potential themes aligning with the project’s social-ecological framework. Themes were clustered further into overarching themes associated with the school built environment and bullying. Themes and subthemes were confirmed by a second researcher to maintain confirmability and enhance dependability [30]. Minor inconsistencies were noted, with differences discussed until a consensus was reached. This paper presents participant comments that link school built environment factors to bullying behaviour. Only recurring themes have been presented.

## 3. Results

### 3.1. Study Sample

Seventy interviews were conducted with policymakers (*n* = 22), school staff (*n* = 12), parents (*n* = 18) and students (*n* = 18). Table 1 describes the participant characteristics of each study sample.

### 3.2. School Built Environment Factors Impacting Students’ Bullying Behaviour

School bullying locations or “hotspots” identified by participants when describing bullying incidents at school included locker areas, toilets, transition spaces (i.e., staircases, corridors, queues), outdoor play spaces (including playgrounds and courts), changerooms, classrooms and school buses. Although many school locations were identified as potential sites of bullying, some locations, such as lockers and toilets, were identified more frequently than others. Built environment factors associated with these school locations and bullying behaviour have been grouped according to their influence on (i) visibility and supervision; (ii) physical and psychological comfort and safety; and (iii) social-emotional competencies.

#### 3.2.1. Built Environment Factors Impacting Visibility and Supervision

The school built environment factors impacting bullying behaviour via visibility and supervision included lighting and windows; school and building design; crowding and school size; and security cameras.

##### Lighting and Windows

Participants indicated that optimising visibility by incorporating artificial lights, glass walls, or windows into school buildings not only allowed bystanders to witness and intervene if bullying incidents occurred but potentially prevented bullying from occurring in those areas in the first instance. Some participants stated that bullying perpetrators avoided bullying because they feared getting caught, or alternatively, they observed school staff role-modelling positive interactions. This opportunity for supervision was reflected in a quote by a policymaker who noted that even the potential for staff supervision could reduce bullying behaviour:


*Supervision definitely does [impact bullying]…we place the teacher’s preparation areas- where their desks are—in the middle of active hubs where the students are during or in-between class time…we have the walls transparent because we want [the teachers] to see outside…we find if teachers aren’t there because the hub is there, students act differently.*
(Policymaker #01)

Many participants also indicated that poor-quality lighting both limited visibility and prompted feelings of discomfort or irritability that potentially impacted bullying behaviour:


*Often you just want to get a bit of natural light into the lockers, and the actual light affects the way people are using the space…people are feeling nervous already when they enter those [dark] spaces….I think [natural light and good ventilation] encourage people to be acting in a more appropriate way.*
(Policymaker #01)

Similarly, the source, location and strength of lighting were said to influence bullying behaviour. While bright lighting was thought to improve visibility, soft lighting was associated with feelings of wellbeing that may prevent bullying:

*I think [preventing bullying] is having lots of lights, lots of windows, lots of natural fresh air…Well laid out lighting systems that are soft, that are in the right places, they’re not so harsh…* promote that feeling of wellbeing…you’ve maybe then got a chance of supporting those that are feeling the need to bully…they’re not going to feel as agitated themselves, or as hostile themselves, or as trapped in the environment.(Parent #07)

##### School and Building Design

Many participants noted that visibility could be compromised by obstructive building configurations and architectural designs, as well as natural elements, such as trees. For example, locker areas were identified as common bullying locations due to design features that limited visibility:


*[Bullying occurs] where locker areas are like rooms or where you don’t have multiple exits…Where they are enclosed and quite dark—that is where people can be doing things around the corner and not be seen.*
(Policymaker #01)

Similarly, students noted that hidden corners within school corridors and buildings were bullying hotspots: *At the high school, there’s these little corners you can go into… students go into that [corner] and fight* (Student #01). Students also noted, however, that obstructive equipment can provide sites for targets of bullying to hide from perpetrators of bullying: *A good spot to hide if you need to is behind the playground because the play equipment blocks it off a bit.* (Student #11)

Unclear pathways that lacked signage and potentially led to hidden or under-utilised areas of the school were also linked to bullying. As one policymaker commented:


*Clarity of order is really important for spaces to discourage bullying…It’s not just about visibility, but knowing that you are going in the right direction; you’re confident and you can clearly see the way from a to b, which again, discourages bullying…even for the bully, when you’re moving from space to space, you have less time to bully and less opportunity to bully.*
(Policymaker #01)

Participants also indicated that locating buildings and services on the perimeter of the school grounds could exacerbate bullying behaviour:


*We’re also concerned about toilets and how they’re very off-to-the-side in some schools… They’re on the periphery…students…go off there unsupervised.*
(Policymaker #15)

The visibility and supervision of students was influenced by policies within the school. For example, some school locations were identified as bullying hotspots because teachers were not permitted to enter those areas, or in some cases, chose not to:


*Toilets and change rooms are another one that present a problem for schools because no teacher wants to go into a student change room where students are undressed…they don’t really want to go into a toilet for the same reason. And yet it’s an area that presents risk because the students know they can’t be supervised…the whole thing is designed to avoid sight lines. Normally you can’t see through the lobby to the actual toilet cubicles.*
(Policymaker #01)

##### Crowding and School Size

Crowded areas, characterised by too many people in the space available, were also perceived by study participants as interfering with visibility and students’ physical comfort and perceived safety:


*If you don’t give [students] enough space, then they are going to fight… it can become bullying. They’re going to fight through the corridor…. they knock shoulders and then all of a sudden you’ve got [an altercation] …in a physical space it’s really important to make sure there is enough room, particularly in the main movement spaces, to encourage nicer feelings [and to ensure students] are interacting in a controlled way.*
(Policymaker #01)

Locker areas were also criticised for being crowded spaces with limited supervision that facilitated bullying behaviour:


*What you don’t want is a big conglomeration of 200 lockers in one small room …the tall lockers that are two metres high, one on top of the other, in rows… [students] are all trying to get in and get out… it’s hard to supervise because the lockers themselves block sight lines.*
(Policymaker #02)

The size or number of students in a school or classroom was also discussed in relation to crowding, with many participants associating larger schools or classes with more bullying behaviour:


*I think one of the main [reasons students who were bullied at other schools moved to our school] would be the size of the school, or maybe the size of the classrooms as well. Some of the [other] schools have quite large classes which makes it harder to keep an eye on things.*
(School staff #07)

##### Security Cameras

Many participants linked the absence of closed-circuit television (CCTV) cameras in schools to poor visibility and increased bullying and vandalism:


*[Bullying] always occurs where there are no cameras…places like your toilets, your shower areas, change rooms; where teachers don’t have access and where there’s no cameras.*
(Parent #07)

Bullying on school ovals was also attributed to a lack of CCTV cameras, with one student noting:


*On the ovals there aren’t much security cameras and it’s a big oval so harder for people to get to.*
(Student #16)

However, despite the proposed relationship between CCTV cameras and bullying by many participants, one policymaker did question whether security cameras simply changed the location of bullying to areas without cameras and encouraged covert rather than overt bullying behaviour.

#### 3.2.2. Built Environment Factors Impacting Physical and Psychological Comfort and Safety

School built environment factors thought to influence physical and psychological comfort and safety included ventilation and temperature; acoustics and noise; queues; aesthetics, vandalism, and maintenance; and furniture and seating.

##### Ventilation and Temperature

Poor ventilation was frequently linked to bullying victimisation in schools, particularly in students experiencing health problems and respiratory conditions:


*There’s a huge connection between health and wellbeing and bullying…You imagine a child has asthma… If kids see someone weak, they pick on them…[Asthmatic students may not be] able to go into a classroom because it’s too dusty.*
(Policymaker #06)

Study participants also linked poor ventilation to unpleasant body odours, which may prompt bullying behaviour:


*Ventilation is terribly important…you’ve got 30 kids running around…they come in a class and they all smell.*
(Policymaker #06)

In addition, participants recognised that some students had underlying health issues and were particularly sensitive to extreme temperatures. Students who had difficulty regulating their temperature were potentially more susceptible to bullying behaviour:


*A lot of kids with poor temperature control visually sweat a lot so bullies really hone in on the fact they can see sweat…and quite often, they will smell, even if they are clean.*
(Parent #07)

Extreme temperatures were also said to make students irritable, potentially triggering bullying perpetration:


*The [outdoor] undercover areas are so cold and grey…children are having to sit and eat their food on a really cold floor…it’s uncomfortable and you tend to see the children getting restless and starting to pick on each other.*
(Policymaker #05)

##### Acoustics and Noise

Participants often noted that poor acoustics, and accompanying loud or irritating noises, can create discomfort among bullying perpetrators, increasing irritability, and prompting bullying incidents:


*[The impact of noise and physical discomfort on bullying] works in two ways: one, if you are the person being bullied, you are already nervous in that space and the bullying will affect you more because you are not feeling safe and secure…then if you are a person who is doing the bullying, already you are grumpy from [the noise and discomfort] and [bullying is] what you are using to sort of relieve that or express it.*
(Policymaker #01)

Participants also commented that built environment strategies used to manage extreme temperatures can inadvertently exacerbate noise pollution:


*You need the air conditioner running quite a lot during the summer period, and it tends to be very noisy in class…that sometimes then makes the children a little bit more agitated…if the students are more agitated, then I think bullying could be more likely to happen, just because they’re getting more worked up or they want to find some way to vent that feeling.*
(School staff #07)

##### Queues

Transition spaces that required students to queue or move slowly, such as staircases and corridors, were frequently identified by study participants as common bullying locations. Many of these “unowned” spaces lacked supervision and were crowded or uncomfortable:


*I think anytime you ask children to queue…you are providing a place that they can nudge and whisper and pass on content about somebody else…In canteens kids are “hangry” so they are wanting food. It’s often time-dependent because equipment and places to sit are social priorities in schools…You want the basketball court first. You want the seat that’s under the shade…so kids want to be at the front of the lines so they can get the seats, so that they are not the last ones arriving…and perhaps there’s not teachers around supervising all of the lines.*
(Policymaker #07)

##### Aesthetics, Vandalism, and Maintenance

The influence of school aesthetics on bullying behaviour was frequently mentioned by study participants, many of whom linked school aesthetics and the related concepts of vandalism and maintenance to students’ attitude towards the school and other students:


*There were some schools where the upkeep was so poor that the kids saw no value for the school and they also saw no value then in the children who were attending it. So [the school] almost encouraged lawlessness. Nobody cares about anything so you can get away with anything…If the culture is one of neglect of the school, then it’s one of neglect of the students as well.*
(Policymaker #09)

Many participants also emphasised the importance of including different spaces in schools to cater to the different needs of students:


*If the school’s not looking like it’s a place that people respect, the students aren’t going to respect it. And that permeates out into then how we respect one another…[a school that looks respected] would be a school that didn’t have graffiti, it would be a school that has lots of beautiful plant life, has lovely spaces for students to be together, and different kinds of spaces. So spaces to run, spaces to play, spaces to sit and have a chat. So it’s very responsive to the needs of the different age groups and the students.*
(Policymaker #18)

##### Furniture and Seating

Participants often noted that the configuration of classroom furniture—particularly desks and seating—influenced students’ comfort levels and response to bullying incidents. Furniture placement also influenced bullying perpetrators access to other students:


*We have coffee tables, cushions, stools, high tables, low tables…and then normal traditional tables and chairs as well. And the kids love it. I can also see that could, in some environments, lead to an increase in bullying in that they’re not just sitting in one desk, one chair for a whole day…because in this way [perpetrators of bullying] have access to everybody within the classroom.*
(School staff #02)

Some participants also noted that neurodiverse students might struggle with unassigned seating plans or desk configurations:


*I think there are two [reasons bullying occurs in the classroom]…when tables are put in groups where children are looking straight at each other, or when there is no assigned seating plan. So, for primary school, that was a really big issue for [my neurodiverse son] because he felt excluded so easily with no assigned seating plan…he watches and perceives people very acutely. When he has people sitting directly facing him, the slightest thing he can find very intimidating or he would think purposeful…So seating, and when they are in a group, he would feel excluded more often because students may say something, one particular word, and he thinks they’re saying something where he’s not invited.*
(Parent #02)

However, some school staff stated they preferred assigned seating plans so they can separate students who may engage in bullying behaviours.

#### 3.2.3. Built Environment Factors Impacting Social-Emotional Competencies

School built environment factors impacting bullying behaviour via social-emotional competencies included: (i) collaborative workspaces; (ii) quiet spaces for reflection and mood regulation; and (iii) signs of inclusivity.

##### Collaborative Workspaces

A common strategy suggested by participants to reduce bullying in schools was the provision of spaces for collaborative group work and the development of skills that can prevent bullying behaviour, such as managing boredom, exclusion, conflict, mood regulation, and discrimination. Collaborative workspaces were described by some participants as school locations with different zones to accommodate different numbers and needs of students. Built environment factors that supported collaboration included open or adjoining classrooms and equipment for group activities. One participant described an open classroom as follows:


*When I’m talking about an open classroom, I’m not talking about a classroom without walls. I’m talking about defining different zones with tables and storage units…where you can meet in smaller groups, or you can meet independently. Giving people choice of where they can sit.*
(Policymaker #04)

A lack of appropriate outdoor furniture and seating in outdoor areas was also linked to opportunities for social interactions that can either prevent or exacerbate bullying behaviour:


*The power of a chair cannot be understated…People often arrange gardens so that they are aesthetically pretty, rather than thinking about the social dynamics of that environment. Straight-line walls are terrible because that encourages all the kids to sit on the wall and judge all those that are going by…a curved wall…encourages conversation…more children are welcomed in.*
(Policymaker #07)

In addition to absent or inappropriate outdoor furniture, many participants noted that inadequate equipment or activities during break times were linked to negative social interactions, and possibly, bullying:


*[Students] that feel left out are the ones that are going to be either picked on because they’re not doing anything, or they’re going to be the ones that are trying to get in and cause those issues in the first place. [Preventing bullying]…is definitely having things for [students] to keep them occupied.*
(Parent #18)

Indeed, one participant commented on an increase in bullying behaviour when some outdoor equipment was removed during the COVID-19 pandemic:


*Particularly over COVID, we put the loose parts away…but we actually noticed without it, the [bullying] behaviour spiked. So then when we brought it out again…we saw that spike dissipate…[loose parts are] just basically any old piece of rubbish…old keyboards, old kettles, all your sand play equipment.*
(School staff #06)

Similarly, restricting students to indoor learning environments was thought to influence bullying behaviour:


*Some children… they’re more hands-on, so they like to be outside, they’re more athletic… they don’t want to be in the classroom. So having that alternative [of working outside] for them might take some of the bullying out of the classroom…My son would get picked on in the classroom…most of the time, it’s from a kid that’s bored, or is not doing their work or just can’t be bothered…So having that child taken out and maybe doing more hands-on activities would be good for them.*
(Parent #17)

##### Quiet Spaces for Reflection and Mood Regulation

Many participants also suggested that creating specific spaces within schools for student reflection and mood regulation may prevent bullying:


*Places need to teach children how to regulate their emotions…children who get really angry, really quickly…are often children who have been victimised themselves…sometimes these self-soothing rooms and reflection rooms are fabulous for children who are reactive bullying or bully-victims.*
(Policymaker #07)

Similarly, participants noted that areas within an existing classroom can be designed to ensure students have spaces for reflection or mood regulation. For example, a cost effective strategy suggested by participants included a portable tent in the corner of a classroom that provided students with a sense of privacy while still connecting them to the larger space and other students.

##### Inclusive Environments

Lastly, the absence of inclusive environments or facilities that reflect an acceptance of marginalised groups within schools was seen to influence bullying. Marginalised groups may include students who are LGBTQIA+, disabled, living in poverty, or members of an ethnic or religious minority group. Some participants noted that creating school environments that accept and celebrate differences between members of the school community may assist students in developing important social-emotional competencies, such as self-awareness, cultural identity, empathy, self-management, and relationship skills. In addition to including gender-neutral toilets in schools to prevent bullying of sexuality and gender-diverse students, study participants noted that signage throughout the school can reflect an inclusive environment:


*If kids don’t feel included then the very processes by which the school is engaging with the community can feel exclusive. For example, [one school] had this big sign up out the front that had ‘welcome to the school’ in most languages of the world…The language that was missing was [local Australian Aboriginal language] Nyoongar.*
(Policymaker #09)

## 4. Discussion

This study explored school policymaker, practitioner, staff, student and parent perspectives of school built environment factors influencing bullying behaviour in primary and secondary schools. Participants rarely linked the school built environment directly to bullying behaviour. Rather, the school built environment was linked indirectly to bullying via three potential mediation pathways: (i) visibility and supervision; (ii) physical and psychological comfort and safety; and (iii) social-emotional competencies. Notably, many built environment factors were thought to influence multiple pathways. For example, study participants linked lighting and crowding to both visibility and comfort and safety. Figure 1 proposes a conceptual model based on these qualitative findings suggesting potential pathways between the school built environment and bullying behaviour.

The proposed pathway linking the school built environment to bullying behaviour via visibility and supervision is reflected in the published literature. The built environment has previously been found to impede student-staff visibility [21,22], while poor staff supervision of students has frequently been associated with increased bullying behaviour [2,23,24,31]. However, less structured school settings that are poorly supervised (e.g., large playgrounds) can also provide students with opportunities to develop social-emotional competencies that may prevent bullying, such as problem-solving and negotiation skills [21,32]. Future studies may consider the type of supervision needed to optimise social and emotional learning while preventing bullying behaviour.

While no studies were found that explored the mediating role of physical and psychological comfort and safety on the relationship between the school built environment and bullying, Lamoreaux and Sulkowski’s (2020) review explored the effects of the school built environment on student safety and psychological wellbeing [33]. Focusing on the crime prevention through environmental design (CPTED) architectural philosophy that promotes the design of built environments to reduce fear and prevent crime, the review identified a number of studies linking the school built environment to physical and psychological comfort and perceived safety [34,35,36]. For example, one study of 900 middle and high school students in the U.S. found that students preferred CPTED-based school designs over non-CPTED schools for their perceived safety and psychological comfort [34]. CPTED-based designs incorporated built environment factors that enhanced natural surveillance (e.g., open layouts, large windows); access control (e.g., locking doors, monitoring visitors); territoriality (e.g., signage and landscaping); and maintenance (e.g., building deterioration). The authors noted that physical safety and psychological comfort were interrelated, with physical safety defined as “feeling protected from crime, bullying attacks, or other physical harm” (p. 481) and psychological comfort as a state of positive well-being and low stress in which students were mentally ready to learn [34]. The built environment has also been linked to physical comfort in studies exploring building performance and occupant satisfaction, functioning and school attendance [37,38]. Comfort variables are often described as Indoor Environmental Quality (IEQ) and include thermal comfort, lighting, humidity, acoustics, and ventilation [37,38]. Noise, lighting, temperature, air quality, and furniture have been linked to mental health and stress in non-school settings [39,40], while school built environment factors linked to perceived safety have included poor supervision, noise, crowding, disorder (e.g., litter, vandalism, graffiti) and security features (e.g., fences, security cameras) [2,17,41,42,43]. Additionally, feeling unsafe at school has been associated with greater involvement in bullying incidents [44,45].

This study found students’ social-emotional competencies may be improved by creating collaborative workspaces and areas for reflection and mood regulation, providing sufficient equipment during break times, and displaying signs of inclusivity within the school. However, improving students’ social-emotional competencies is not sufficient on its own to reduce bullying behaviour [13,14]. Rather, social and emotional learning needs to be incorporated into comprehensive whole-school anti-bullying programs [14]. The finding that diverse spaces and sufficient equipment in schools may help develop students’ social-emotional competencies and, in turn, prevent bullying behaviour is reflected in other studies showing busy playgrounds lacking peaceful spaces and adequate equipment can incite negative interactions, including bullying, between students [22]. Similarly, larger, less densely populated spaces have been shown to facilitate collaborative play [22]. However, our findings also highlight the challenges faced by school staff and architects when designing schools catering to students with diverse needs and preferences. For example, versatile seating and desk arrangements were associated with both the prevention and exacerbation of bullying behaviour in classrooms. Operable windows were said to improve ventilation and indoor air quality in unpolluted regions, however, they can also interfere with acoustic comfort [46]. The challenge of finding a classroom temperature that suits all students was also reflected in the finding that some health conditions can limit students’ ability to regulate internal body temperatures, while study participants also discussed different beliefs about the benefits of CCTV cameras. Indeed, some studies have indicated CCTV cameras are highly effective anti-bullying interventions [47], while others found cameras were associated with higher odds of fearing victimization [48]. Further research is needed to determine the impact of CCTV cameras on bullying behaviour.

A limitation of this study is the potential over-representation of participants who experienced bullying victimisation rather than perpetration. While school coordinators were asked to nominate students and parents with different experiences of bullying, few students or parents identified themselves or their children as perpetrators of bullying incidents during the interviews. It is unclear if this was because fewer students who perpetrate bullying participated in the study or if they chose not to divulge this information. Similarly, all nominated parents were female. Researchers may need to work more closely with schools to ensure a broader range of participant perspectives, particularly perpetrators of bullying and male parents or carers. Nonetheless, this study involved a diverse mix of participants from different school sectors, primary and secondary schools, low and high socio-economic areas, and old and new school builds. While this study has identified potential relationships between school built environment factors and bullying behaviour, including potential mediators of this relationship, it is not an exhaustive list of the potential pathways or mediators linking the school built environment to bullying. Future studies are needed to test the pathways proposed in this study and explore other potential mediators commonly associated with bullying behaviour, such as access to vulnerable students, power, aggression, and school culture.

School strategies to support minority groups and vulnerable students also merits further investigation. For example, neurodiverse students were frequently mentioned by study participants, particularly during discussions about classroom seating, furniture, and break-time activities. Given the increased rate of neurodevelopmental diagnoses, such as autism, in Australian school children, future research might explore how the school built environment impacts bullying behaviour involving neurodiverse students [49]. Similarly, toilet and locker areas were identified as bullying hotspots by many participants, with toilets often linked to the bullying of sexuality and gender-diverse students. While the authors have recently explored barriers to including gender-neutral toilets in schools [50], future studies could explore the optimal design of these spaces for all students.

The identification of pathways through which school built environment factors impact bullying behaviour has a number of implications for policy and practice. For example, visibility and supervision may be improved by ensuring schools are designed to be well-lit with sufficient windows, clear sightlines, appropriately sized spaces, and security cameras. Physical and psychological comfort and safety may be improved by ensuring the effective management of ventilation, temperature, and acoustics. Queues and transition spaces also need to be well-managed, and school facilities, furniture and equipment should be well-maintained. Students’ social-emotional competencies may be improved by designing collaborative workspaces containing adequate equipment, spaces for reflection and mood regulation, and displays of inclusivity. As such, school architects are crucial collaborators when designing anti-bullying interventions. Schools should also be co-designed with policymakers, school staff, students, and parents. In particular, students’ voices should be prioritised when identifying bullying hotspots and school built environment factors influencing students’ perceptions and experiences of safety and bullying behaviour. System and school-level leadership teams need to ensure time and resources are available to school staff to identify and address changes to the school built environment that may prevent bullying behaviour. Fram and Dickmann (2012) noted that preventative changes to the school built environment to reduce bullying behaviour should be preceded by the collection of bullying data from staff and students, as well as physical scans of buildings [21]. School-specific data is crucial to ensuring built environment interventions target the unique needs and contexts of each school. School staff and community involvement and commitment to the implementation of anti-bullying interventions will help to ensure efforts to prevent bullying behaviour are effective and sustained [21].

## 5. Conclusions

This study explored school policymaker, practitioner, staff, student and parent perspectives of school built environment factors influencing bullying behaviour in Australian school students. The school built environment has the potential to impact bullying by influencing visibility and supervision, physical and psychological comfort and safety, and social-emotional competencies. Ensuring school built environments are designed to reduce bullying behaviour in school students may be an effective population level health intervention to support bullying prevention efforts in primary and secondary schools.

## Figures and Tables

**Figure 1 ijerph-19-15955-f001:**
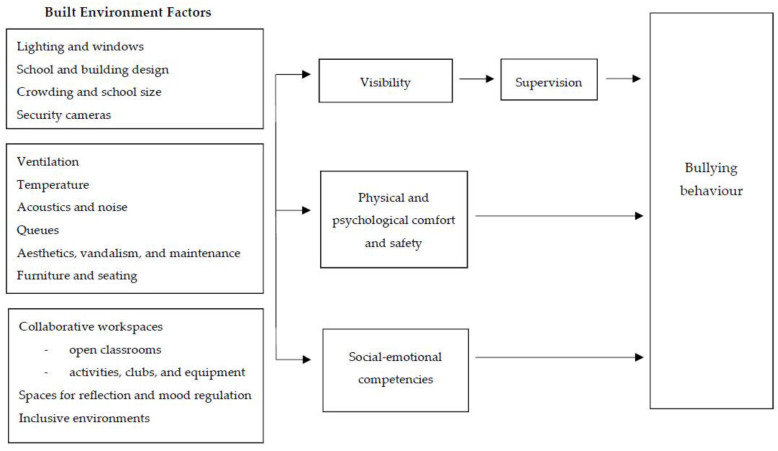
Potential pathways linking school built environment factors to bullying behaviour in school students.

**Table 1 ijerph-19-15955-t001:** Participant characteristics.

Characteristic	Policymakers (*n* = 22)		School staff (*n* = 12)		Students (*n* = 18)		Parents (*n* = 18)	
	*n*	%	*n*	%	*n*	%	*n*	%
**Age (years)**								
6–10	-	-	-	-	1	5.6	-	-
11–15	-	-	-	-	16	88.9	-	-
16–20	-	-	-	-	1	5.6	-	-
21–30	-	-	-	-	-	-	-	-
31–40	2	9.1	6	50	-	-	6	33.3
41–50	8	36.4	9	75	-	-	9	50
51–60	6	27.3	3	25	-	-	3	16.7
61–70	6	27.3	-	-	-	-	-	-
Mean	53.3	-	47.2	-	13	-	45.5	-
SD	8.7	-	7.7	-	1.7	-	6.5	-
**Gender**								
Male	10	45.5	5	41.7	11	61.1	-	-
Female	12	54.6	7	58.3	6	33.3	18	100
Prefer not to say	-	-	-	-	1	5.6	-	-
**Born in Australia**								
Yes	15	68.2	9	75	17	94.4	16	88.9
No	7	31.8	3	25	1	5.6	2	11.1
**Marital status**								
Married	-	-	-	-	-	-	14	77.8
De-facto	-	-	-	-	-	-	3	16.7
Single	-	-	-	-	-	-	1	5.6
**Years in role**								
1–10	2	9.1	3	25.0	-	-	-	-
11–20	8	36.4	5	41.7	-	-	-	-
21–30	9	40.9	3	25.0	-	-	-	-
31–40	2	9.1	1	8.4	-	-	-	-
41–50	1	4.5	-	-	-	-	-	-
Mean	23.6	-	9	-	-	-	-	-
SD	18.1	-	9.1	-	-	-	-	-
**School grade**								
Grades 4–6	-	-	-	-	6	33.3	-	-
Grades 7–8	-	-	-	-	6	33.3	-	-
Grades 9–10	-	-	-	-	6	33.3	-	-

## Data Availability

Data are available upon reasonable request from the first author.

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
