# Peer review of "School Built Environments and Bullying Behaviour: A Conceptual Model Based on Qualitative Interviews"

_ijerph, 2022, doi:10.3390/ijerph192315955_

Round 1

Reviewer 1 Report

This paper sets out to explore something important, namely how the school built environment impacts bullying behaviour. I genuinely liked this paper and I think that for most parts it is a good and solid paper. I therefore only have a few comments that can improve the paper before publication.

Materials and methods

2.1.2.

- A few more details are needed. For instance, what type of interviews were gathered with students, individual?

2.2  

- Could some examples of questions asked be provided? I also wonder about the fact that different guides were used and whether they targeted the same issues? Maybe discuss this in limitation.

- Briefly mention how participants referred to maps during the interviews. What type of maps? How were maps used?

Results

The introduction of the findings describing hotspots is a little confusing as it has not previously been mentioned that participants were advised to talk about that. Perhaps this will be clearer when giving more details around topics/themes/maps in the methods section. Perhaps also give some sort of overview of themes. For example, describe what each theme contains /subthemes.

I also wonder whether the themes identified were shared among the different group of participants, especially I wonder if they were raised also by students? This is of particular importance as only one student excerpt are used in the results. I think more examples from students would be good. If by chance these themes were not raised so much by students, perhaps it might be easy to cut students out and make a paper with full attention on their perspectives instead.

Theme 3.2.3 Inclusive environments

It is not so clear in the results how this theme relates to social emotional competencies. It gets a bit clearer in the discussion so perhaps just a little clarification is needed.  

Limitations

If there were differences in what themes were raised by different groups of participants than these might be raised for discussion here. Likewise, the issue of different guides and if that might have affected the findings in some way.

Reviewer 2 Report

Dear Authors

It was pleasure to read this paper. Research topic is relevant and important for modern education and related sciences and practices.

The abstract is well written, however, the necessity of presenting the dates of the study is to be discussed.

The introductory part of the article lacks an analysis of more recent research.

The methodological part requires addition. As this is work with a qualitative methodology, the study lacks philosophical and epistemological position. And such a part as validity/trustworthiness is not given at all.

I would discuss whether table 1 does not overload the text (line 194). Also I would also suggest adjusting the presentation of the results. So far, it seems that the authors presented only preliminary results in this part, while the disclosure of connections between topics is missing. 

The discussion part is well written, but I would suggest using more recent scientific literature to analyze the results.

The conclusions are only partially related to the research.

Reviewer 3 Report

Dear authors,

congratulation for the research and the paper. I found it very interesting, good introduction, interesting data collection through interviews, and very relevant results regarding how the building could increase or decrease bullying behaviour.

The model presented in fig 1 I see it as very appropriate considering the importance of feeling well in the class and in the school as a way of preventing misbehaviour. Please check the numbers in the figure as probably are a mistake.

Thanks for giving me the opportunity to review the paper.

Round 2

Reviewer 2 Report

Dear authors,

Thank you for taking certain observations into account